# Evaluating the accuracy of CAD/CAM optimized stones compared to conventional type IV stones

Gülsüm Ceylan[1]*, Faruk Emir[2]

1 Department of Prosthodontics, School of Dentistry, Istanbul Medipol University, Istanbul, Turkey,
2 Department of Prosthodontics, Gülhane Faculty of Dentistry, Health Sciences University, Ankara, Turkey

☯ These authors contributed equally to this work.
* gulsumcyln@gmail.com

**Data Availability Statement:** All relevant data are within the paper and its Supporting Information files.

**Funding:** The authors received no specific funding for this work.

## Abstract

This study compared the accuracy (trueness and precision) of stone models fabricated using two brands of CAD/CAM optimized stones Cerec Stone (BC) and Elite Master (EM), and a conventional type IV stone Elite Rock Fast (ERF). 30 conventional Type IV and scannable stone complete-arch models were scanned with a blue LED extraoral scanner, and root mean square values were obtained. 6 abutments were used in complete-arch models. The digital models were compared with the master model to evaluate their trueness using model superimposition with Geomagic software. Precision was determined for each case by superimposing combinations of the 10 datasets in each group. The point cloud density of each model was calculated with MeshLab software. Kruskal-Wallis and Mann-Whitney non-parametric tests were used for the statistical analysis. The trueness of the stone models was 96 μm for the BC, 88.2 μm for the EM, and 87.6 μm for the ERF. There were no significant differences between the tested dental stones (p = .768). However, the EM models (35.6 μm) were more precise than the BC (46.9 μm) and ERF (56.4 μm) models (p = .001, p < .001). EM models also showed the highest point cloud density. There were significant differences in point cloud density (p = .003). The EM models showed significant differences in precision but no significant differences in terms of trueness. Although EM was more precise and had the highest point cloud density, all models were within the clinically acceptable limit.

## Introduction

Despite the recent advances in CAD/CAM technology, stone casts are remain an essential part of laboratory procedures in dentistry, such as fabricating working and diagnostic models, removable dies, mounting casts, and porcelain layering [1–3].

Accuracy is crucial in fixed restorations' marginal and internal fit. Therefore, accurate replicas of teeth are essential to obtain clinically acceptable results [4]. Dental stones are cheap, easy to handle, widely available, and compatible with impression materials [2, 5, 6].

According to American Dental Association (ADA), gypsum products are classified into five types, from type I to type V [7]. Type IV dental stone has high strength and low expansion,

**Competing interests:** The authors have declared that no competing interests exist.

and is used to fabricate definitive casts and dies for a long time [8]. A void-free definitive cast is fundamental in dental technologies, including CAD/CAM and digital scanning systems. Since the development of intraoral scanning is still ongoing to optimize accuracy and time-cost issues, conventional impressions and extraoral scanning of stone models are widely used [1]. According to International Organization for Standardization (ISO) No. 6873, scannable stones should have high dimensional stability and reproducibility and low expansion (0.07%), and minimal bubble formation [7]. The surface of scannable stones should not be shiny, should not require powder during scanning, and should have a very smooth surface and low surface roughness for superior scans [9].

Different devices such as analog or digital calipers, profile projectors, and microscopes have been to evaluate the accuracy of stone casts. These devices allow for the measurement of linear distances between specific landmarks [10]. In recent years, developments in digitizing systems have given us the ability to perform 3-dimensional measurements on virtual models. This has allowed us to make a more quantitative and qualitative analysis of the accuracy of indirect restorations and to have better information [11].

Accuracy is comprised of the following parameters: trueness and precision [12, 13]. Trueness is the deviation of the fabricated object from its actual dimensions [12, 14]. Precision is the deviation between repeated measurements [13]. The stone model and removable die materials should be accurately scannable because the stone cast material can affect the precision of data acquisition during scanning [9].

Recent advances in CAD/CAM technology have led to the development of scannable dental stones. According to manufacturers, scannable stones are specially formulated to be optimized for use with CAD/CAM systems and are available in a range of colors that are easily detected by modern laser and optical scanners [15]. Despite their potential benefits, there is still a lack of data on the accuracy of scannable dental stones. Therefore, comparing scannable dental stones to traditional dental stones is necessary to determine their effectiveness during extraoral scanning procedures [1]. In this study, two types of CAD/CAM optimized stones and one type IV stone, commonly used in dental clinical settings, were selected. The aim of this study was to evaluate the accuracy of working casts fabricated from CAD/CAM optimized stones in terms of trueness and precision. The null hypothesis was that there are no statistically significant differences between the tested materials.

## Materials and methods

An arch-shaped master model measuring 14 mm in height and 16 mm in width was designed using CAD software (RapidForm XOR2; 3D Systems). Six abutments representing prepared teeth (right and left mandibular second molar, right and left mandibular second premolar, right and left mandibular canine) with a height of 10.15 mm and 6° total angle of convergence with 1mm shoulder finish line were placed on the arch. The abutments were designed according to ANSI /ADA specifications, which are also similar to abutments on the test model used in ISO 12836:2015 specification (Digitizing devices for CAD/CAM systems for indirect dental restorations-Test methods for assessing accuracy) [16].

The digital master model was then saved in Standard Tessellation Language (.stl) format and printed using a Polyjet 3D printer (Objet30 Prime, Stratasys Ltd.). The Polyjet 3D printers utilize materials that are extruded from nozzles or a photopolymer that is jetted over the workspace. Then the object is solidified through polymerization with the use of a UV light source [17]. The Objet30 Prime printer uses the intuitive "Objet Studio" 3D printing software. A rigid and transparent photopolymer material named VeroClear was used to print the master model, while the SUP705 material was used for support structures. The SUP705 material is removable

with a waterjet, so no post-curing process was necessary. The production time of the master model took 4 hours and 10 minutes.

An industrial structured blue LED light 3D scanner (ATOS Core 200 5M, GOM GmbH, Braunschweig, Germany) was selected for use. The scanner was calibrated and tested according to VDI/VDIE 2634 (VDI e.V.; Düsseldorf, Germany), displaying maximum deviations: 2 µm probing error form (sigma), 4 µm probing error (size), 7 µm sphere spacing error and 8 µm length measurement error. The printed master model was then scanned ten times with the ATOS scanner and the scan data was merged into a single file using computer software. The data was then exported in STL format.

Polyvinyl siloxane (PVS) impression material and a one-step impression technique were used for the impression procedure. The impression materials were mixed according to the manufacturer's instructions. The PVS putty impression material (Vinlybest; BMS DENTAL, Capannoli, Italy) was hand-mixed until a homogeneous mixture was obtained within 30 seconds, and it was then inserted into the customized tray. The light-body material (Vinlylight; BMS DENTAL, Capannoli, Italy) was simultaneously spread on the master model. The impressions were allowed to polymerize, and the trays were removed after the materials had set. A total of 30 impressions were made under the same room conditions and stored at room temperature (22° C) for 2 hours before the pouring procedure. All impressions were examined visually, and impressions with voids were excluded from the study.

The dental gypsum products used in this study included two CAD/CAM optimized stones (CEREC Stone BC, Sirona Dental Systems GmbH Bensheim, Germany) (BC) and (Elite Master, Zhermack S.p.A, Italy) (EM) and one conventional type IV stone (Elite Rock Fast, Zhermack S.p.A, Italy) (ERF) (Fig 1). The composition and characteristics of the materials are summarized in Table 1. All dental stones were mixed according to the respective manufacturer's recommendations. Deionized water was used. Each stone cast was poured under vibration (Degussa Vibrator R2; Degussa AG, Germany) and allowed to set for 2 hours at room temperature. A total of 30 stone casts were made (n = 10). Then, all stone models were scanned with the Activity 885 blue light scanner (Smart Optics; Bauman Sensortechnik GmbH, Germany, the accuracy of 6 µm according to DIN ISO 12836) [18], and digital models were obtained.

In the 3D analysis (Geomagic Control, 3D Systems) procedure, a sample size of 15,000 points with a tolerance of 0.001 mm and the best-fit alignment method was used. The 3D analysis software gave the root mean square (RMS) and average maximum and minimum values. Trueness was determined by superimposing the digital master model data on the digital models. Precision was assessed for each digital model by overlaying combinations of the 10 datasets in each group.

The color-coded maps of deviations were created using scans of the stone models. The maps were presented using 3D analysis software. Yellow to red fields represent enlargements, while turquoise to dark blue fields represent contractions on the digital models (Fig 2). The point cloud density of each model was calculated using MeshLab software (ISTI—CNR, Pisa, Italy) (Fig 3). The number of all points that construct the complete digital model was divided by the model's surface area in $mm^2$.

Statistical analysis was performed with a significance level of 95% and %99 using software NCSS (Number Cruncher Statistical System) 2007 (Kaysville, Utah, USA). The distribution of study data was evaluated with the Shapiro-Wilk Test. Since the assumption of normality was not met, Kruskal Wallis and Mann-Whitney non-parametric tests were used to compare the different dental stones and the statistical analysis of the measurements.

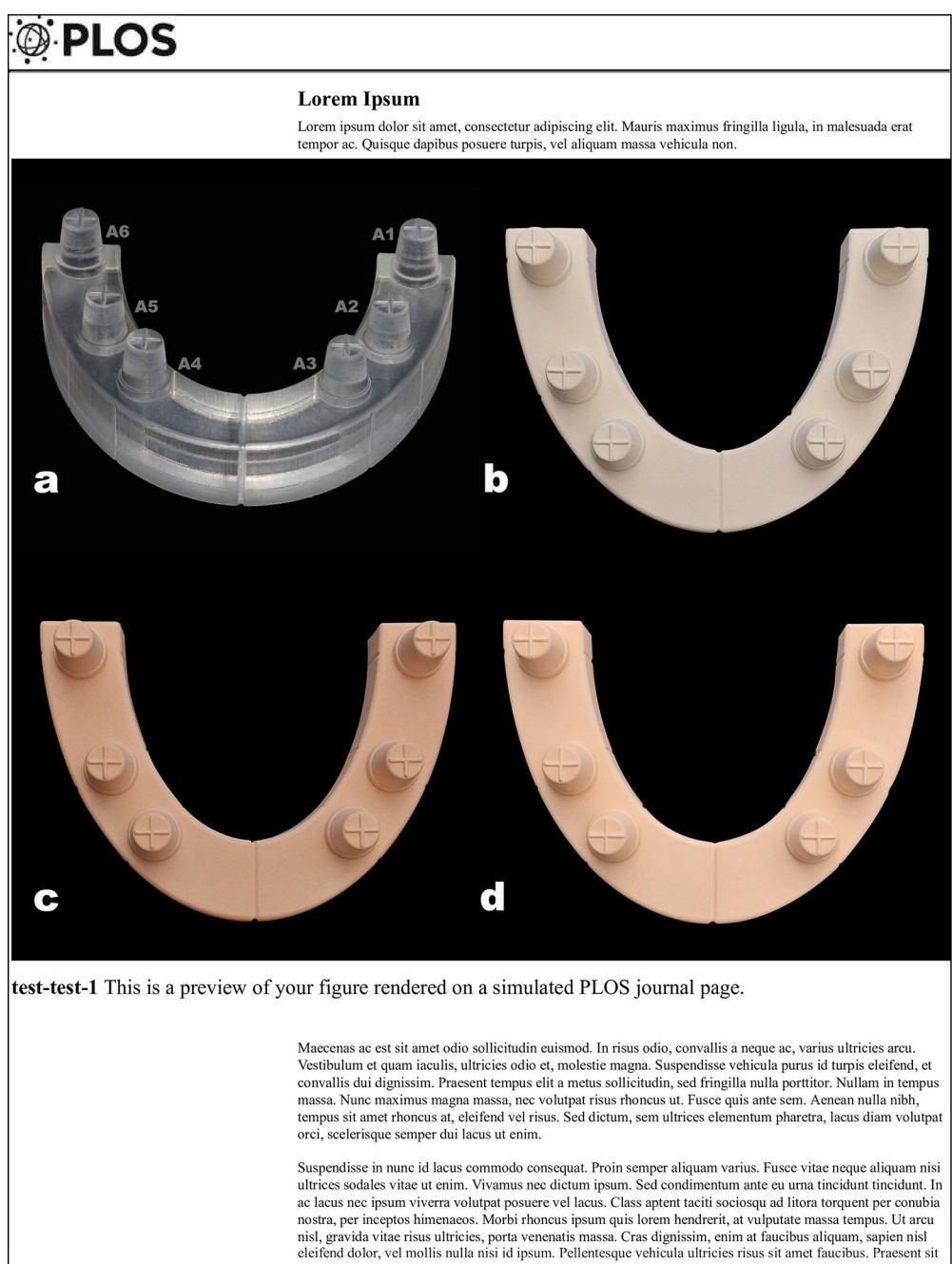

**Fig 1.** **(a)** 3D printed master model and abbreviations for each abutment, **(b)** Stone model fabricated with BC stone, **(c)** Stone model fabricated with EM resin-reinforced stone, **(d)** Stone model fabricated with conventional type IV ERF stone.

## Results

In terms of trueness, the lowest deviation was observed in ERF (87.6 µm), while the highest was in BC (96 µm). However, no significant differences were found among the tested materials (p = .768) (Table 2).

**Table 1. Details and material characteristics of tested gypsum products.**

| Product | Type | Manufacturer | Color | Water/Powder ratio (ml/g) | Setting expansion | Code |
|---|---|---|---|---|---|---|
| **CEREC Stone** | IV (scannable) | Sirona Dental Systems GmbH, Bensheim, Germany | Ivory | 20/100 | <0.08% | BC |
| **Elite Master** | IV Resin reinforced (scannable) | Zhermack, Badia Polesine (RO), Italy | Sandy brown | 21/100 | 0.08% | EM |
| **Elite Rock Fast** | IV (Conventional) | Zhermack, Badia Polesine (RO), Italy | Sandy brown | 20/100 | 0.08% | ERF |

There were significant differences in the mean RMS values of precision of stone models (p = .001). The mean precision was 46.9 μm for the BC models, 35.6 μm for the EM models, and 56.4 μm for the ERF models (Table 3). EM models were more precise than BC and ERF models (p = .001, p < .001).

The comparison of point cloud density measurements among the three stone models showed significant differences (p = .003). The mean point cloud density was 183.3 points/mm$^2$ for BC models, 189.2 points/mm$^2$ for EM models, and 185.1 points/mm$^2$ for ERF models (Table 4). EM models had the highest point cloud density compared to BC and ERF (p = .001, p = .001).

According to group comparison results, significant differences were observed at abutments A1, A2, A3, and A5 (p = .023, p = .026, p = .020, p = .004). For A1, A2, A3, and A5, EM models had the highest RMS values and these differences were significant when compared to BC and ERF (p = .001, p = .001, p = .001, p = .001). However, no significant differences were found for A4 and A6 groups (p = .052, p = .272) (Table 5).

## Discussion

According to the results of the present study, tested gypsum products showed no significant differences in terms of trueness (p = .768). However, EM showed higher precision than BC and ERF (p = .001, p < .001). Therefore, the null hypothesis of the study was partially accepted.

Stone models are considered the gold standard and a fundamental material for diagnosis, treatment planning, and fabrication of prostheses [1, 6, 14, 19–21]. The type IV dental stone is commonly used to fabricate definitive casts and removable dies for fixed prostheses due to its superior mechanical properties, such as high compressive strength, hardness, and low expansion [9, 22–24].

Various impression materials and techniques, different brands of dental stones, and trays are used to fabricate definitive casts, and these factors can affect the accuracy of definitive casts. The definitive casts are scanned by extraoral or intraoral scanners in the digital workflow. Therefore, scanning accuracy is an essential factor that plays a critical role in the long-term success of prosthetic restorations [10, 25]. Different devices such as analog or digital calipers, profile projectors, and microscopes have been to evaluate the accuracy of stone casts [26, 27]. Some studies have reported that the accuracy of intraoral scans decreases with the increased length of the scanned model, which can result in registration problems of the overlapping images [14, 28–34].

Digitizing definitive casts with extraoral scanners in the laboratory is the most common method of CAD/CAM in the dental field. Although using an extraoral scanner adds digitalization errors to the conventional fabrication, this method is still the most reliable option for the production of prosthodontic treatments and shows excellent long-term results [20, 35].

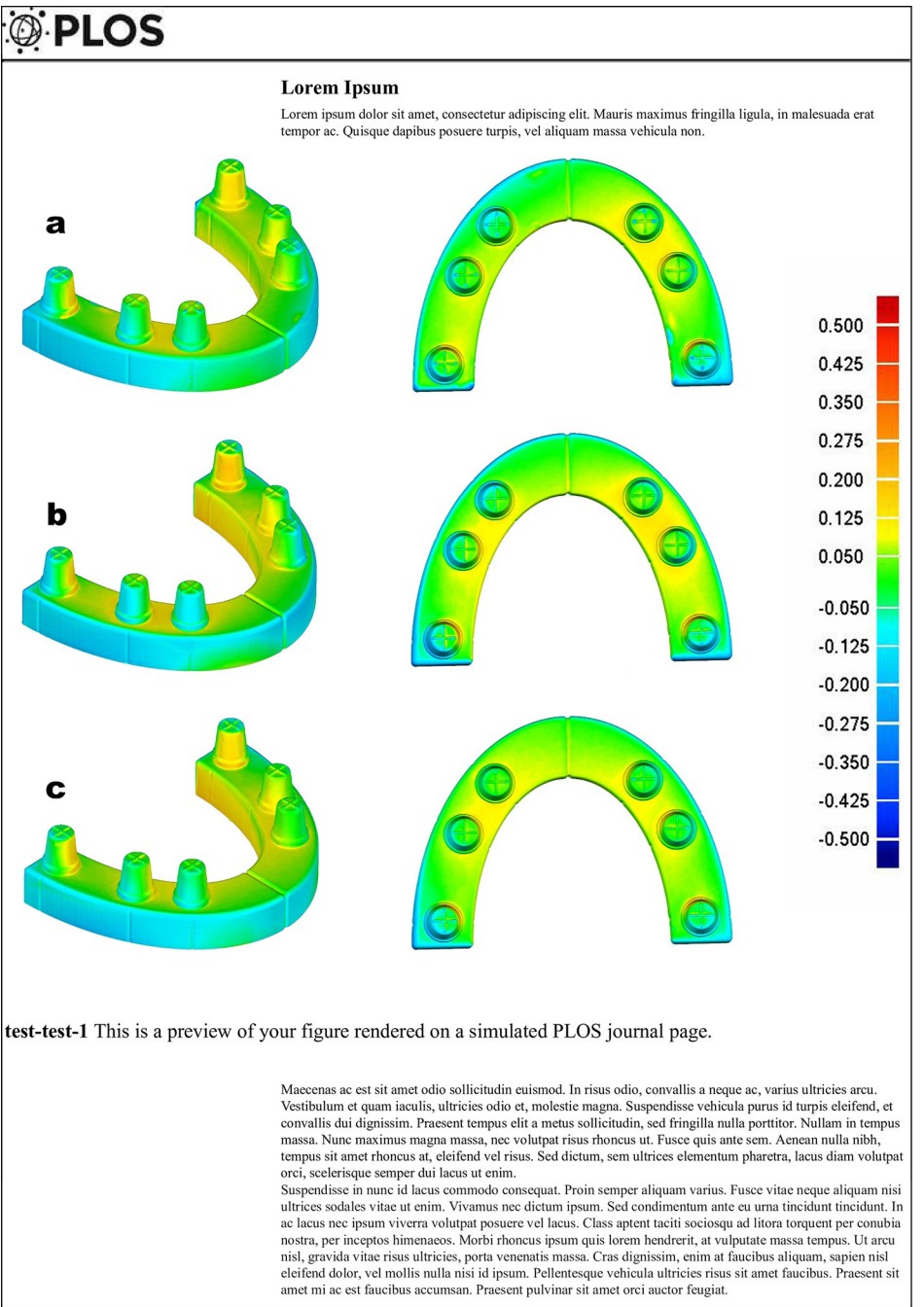

**Fig 2. Typical deviation pattern between tested stone models and master model (trueness).** The deviation range is color-coded from +500 μm (dark red) to −500 μm (dark blue). Maximum/minimum nominal ±50 μm (green). Yellow through red color code indicates that, stone model is larger than the master model; light blue through dark blue color code indicates that, stone model is smaller than the master model; green surfaces present deviations ranging between 0 to ±50 μm. **(a)** BC, **(b)** EM, **(c)** ERF.

Extraoral scanners capture the whole surface to acquire more data simultaneously The blue light-emitting diode (LED) extraoral scanner was used in the present study. The scanner uses stripe light triangulation, and the accuracy was 6 μm [14, 30–32].

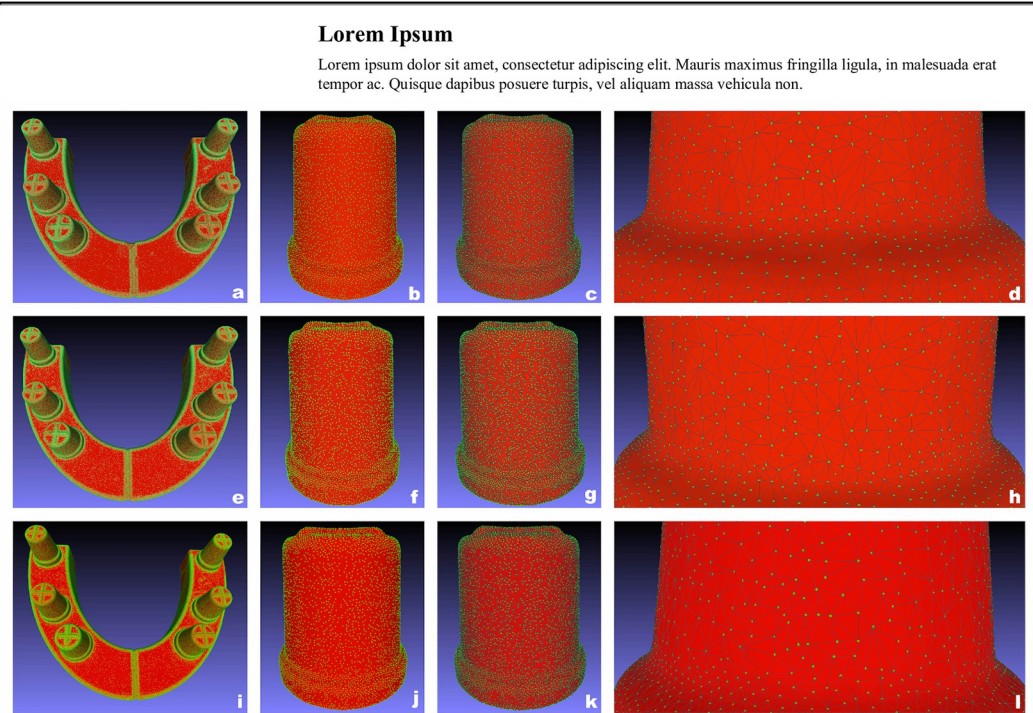

**Fig 3.** **(a-e-i)** Point cloud of each complete-arch stone model. **(b-f-j)** point density shown on each abutment. **(c-g-k)** point density and polygon meshes shown on each abutment. **(d-h-l)** Close view of points and polygon meshes for the comparison of each stone model.

In the data acquisition process, the scanner software generates a point cloud and uses a triangulation process to combine the points and embody the scanned object. The scanner collects information about the distance of each point from the object's surface in every scan. Acquired data is then transferred to a standard coordinate reference system to perform the alignment that combines the points to obtain a complete 3D model of the object [36, 37]. In this process,

**Table 2. Trueness RMS values (µm) for all stone models.**

|  | Mean | Std. deviation | Min | Max |
|---|---|---|---|---|
| BC | 96 | 28.56 | 64.9 | 129.9 |
| EM | 88.2 | 16.48 | 53.9 | 110.2 |
| ERF | 87.6 | 33.78 | 53.9 | 141.6 |

**Table 3. Precision values (µm) for all stone models.**

|  | Mean | Std. deviation | Min | Max |
|---|---|---|---|---|
| BC | 46.9 | 21.05 | 17 | 75.3 |
| EM | 35.6 | 13.84 | 19.4 | 77.8 |
| ERF | 56.4 | 25.88 | 18.2 | 119.3 |

**Table 4. Point cloud density (points/mm$^2$) of all stone models.**

|  | Mean | Std. deviation | Min | Max |
|---|---|---|---|---|
| BC | 183.3 | 2.48 | 178.7 | 187.1 |
| EM | 189.2 | 3.91 | 180 | 195 |
| ERF | 185.1 | 2.13 | 183.1 | 190.7 |

errors due to the point cloud generation during scanning may occur. An increase in the number of points captured during the digitization process lowers errors due to greater data acquisition [38]. However, no correlation was found between the triangle count and accuracy [39, 40]. The accuracy of the digital models relies on the quality of the point cloud created by the software algorithm [40].

**Table 5. Deviations (µm) occurred at abutments A1 to A6 for all groups ($^*$p < .05, $^{**}$p < .01).**

|  |  | Mean | Std. deviation | Min | Max | p |
|---|---|---|---|---|---|---|
| A1 | BC | 23.2 | 3.74 | 20.1 | 30.5 | .023$^*$ |
|  | EM | 30.7 | 7.39 | 18.6 | 41.3 |  |
|  | ERF | 23.3 | 4.69 | 17.7 | 32.4 |  |
| A2 | BC | 22.5 | 3.76 | 17.7 | 29 | .026$^*$ |
|  | EM | 27 | 4.41 | 18.9 | 34.2 |  |
|  | ERF | 22.3 | 3.35 | 14.6 | 26.2 |  |
| A3 | BC | 25.7 | 2.41 | 21.9 | 29.2 | .020$^*$ |
|  | EM | 29 | 2.7 | 25 | 33.3 |  |
|  | ERF | 24.5 | 4.43 | 17.7 | 31.2 |  |
| A4 | BC | 23.3 | 4.03 | 17.8 | 32.4 | .052 |
|  | EM | 27 | 3.73 | 22.3 | 33.8 |  |
|  | ERF | 23.5 | 2.89 | 18.7 | 27.4 |  |
| A5 | BC | 22.9 | 2.02 | 20.8 | 27.4 | .004$^{**}$ |
|  | EM | 28.1 | 4.12 | 24.1 | 35.3 |  |
|  | ERF | 24.1 | 2.14 | 20.3 | 27.6 |  |
| A6 | BC | 22.9 | 3.44 | 16.4 | 29.2 | .272 |
|  | EM | 27.9 | 6.97 | 20.2 | 40.7 |  |
|  | ERF | 24.5 | 3.72 | 19.8 | 30.6 |  |

In order to observe the effect of CAD/CAM optimized stones on the accuracy, the point cloud density of gathered data from the scanned models fabricated with different stones was compared. The number of all points that construct the whole digital model was divided by the model's surface area in $mm^2$. This process gave us the density of data acquired by the scanner. Besides, the improved characteristics of dental stones, such as surface roughness or color, might enable scanners to collect more information. Thongma-Eng et al. [41] reported that the color of the scanned object influenced the accuracy of the scans. Therefore, the colors of the plaster models were selected to be similar to each other. The point cloud densities of BC, EM, and ERF models were 183.3, 189.2, and 185.1 point/$mm^2$, respectively. This difference (higher point cloud density of EM) was statistically significant (p = .003). The density of the points was expected to be different if the CAD/CAM optimized stones were distinctive. Since the same impression procedure and scanner were used for all models, the differences in point cloud density may be related to the material's surface properties, which can affect light reflection characteristics and the data acquisition process of the scanner [9]. In the previous studies, the accuracy of definitive casts was assessed by linear measurements between specific landmarks [10, 42–44]. However, the linear assessment technique limits the evaluation of three-dimensional distortion of definitive casts, and manual measurements may be susceptible to operator performance [3, 10, 19].

Contrary to linear assessment, computer-aided measurements are more reliable and provide advantages in determining the 3-dimensional changes over the complete arch. In addition, dimensional differences can be detected objectively, and these changes can be observed on the color-coded maps [19]. Computer-aided measurement was used as a reference method in some recent studies for high precision analyses [45, 46].

In this method, the software performs best-fit alignment and superimposition procedures. The software reports showed deviation percentages, changes in all directions, average positive and negative deviations, and root means square (RMS) of deviations. If the positive and negative distortions show an equal distribution in the quantitative inspection, average values will be close to zero. For this reason, RMS values were preferred instead of average (+) and average (-) values to evaluate accuracy in the present study [3, 39, 47].

Some studies showed that the accuracy decreases when the scanned area increases [47, 48]. When scanning larger areas, multiple images are merged, and this may lead to progressive distortion and higher inaccuracy [14]. Cross marks on the occlusal surface of abutments and reference lines on the buccal and lingual surface of the arch were added to optimize the accuracy of superimposition process. Therefore, the arch's horizontal and vertical lengths were designed differently to perform the superimposition process effectively. In addition, Seo et al. [10] concluded that the best-fit alignment method might include superimposition errors on complete arches compared to one quadrant. Although the master model in the present study had specific landmarks, each abutment was isolated and analyzed separately to eliminate the influence of deviations associated with arch distortion due to long-span scanning.

Another factor that affects prosthetic restorations' long-term success is the marginal fit. To achieve this goal, internal and marginal gaps have to be set minimum because misfits can jeopardize abutment teeth and periodontal tissues [49, 50].

There is no consensus on acceptable misfit values for prosthetic restorations, but values in the range of 100–150 microns are generally considered clinically acceptable [51–54]. Therefore, the accuracy of stone models used in prosthetic restorations must be within this range or even better. Previous studies have shown that the trueness of complete-arch dental models ranges from 11 to 312 μm [12, 46, 48, 55–58]. In the present study, the mean trueness of the complete-arch model ranged from 87.6 to 96 μm, and the differences were not statistically significant. Cho et al. [48] reported a mean precision of 54 μm for complete-arch casts. Another

study found that the precision of full-arch dental models was 61.3 μm [57]. This study showed that the mean precision of complete-arch models was 35.6 to 56.4 μm. This significant difference (p = .001) in precision values of the EM models might be related to the use of resin-reinforced stone. Kim et al. [9] found that the surface roughness of the resin-reinforced stones is lower than that of conventional type IV stones. Additionally, the physical and chemical characteristics of CAD/CAM optimized stones can affect the resolution and quality of the point cloud data.

Objects with smaller dimensions can be digitized more accurately than larger ones, such as the complete dental arch [12, 34]. Jeon et al. [56] used RMS values to evaluate the trueness and precision of a single master die and found that the mean trueness was 17.4 μm, and the mean precision was. 14.6 μm.

The trueness value of each abutment was decreased compared to the full-arch trueness values and ranged from 22.3 to 30.7 μm. While some findings of other studies are lower than the results in this study, the previous studies used various impression and gypsum materials, master models with different geometries, scanners, and different software programs, which may have affected the outcome and limited the comparability of the results. Additionally, the accuracy of all models made with tested dental stones was within the clinically acceptable limit and could be used to fabricate prosthetic restorations.

Color deviation maps were used in previous studies to reveal the amount and pattern of deviations on scanned models. According to the color-coded maps, deviation patterns were observed similarly for all stone models. The most significant dimensional changes occurred in the posterior region (abutments A1 and A6), where molars were placed. Abutments A1, A2, A5, A6, and the posterior buccal surface of the arch were mostly smaller than the master model. However, the upper surface of the arch displayed a homogenous pattern of green surfaces. Between the abutments A2-A3 and A4-A5, a slight expansion was observed. On the other hand, a slight contraction was observed on the anterior buccal surface of the arch, and a slight expansion was observed on the posterior lingual side of the arch.

The limitations of the present study include: the 3D analysis was conducted in vitro using standardized models, only one impression material and dental scanner were utilized, and the deviations in the X, Y, and Z directions were not determined. Future studies should include the surface roughness of the stone models and different brands of CAD/CAM optimized stones.

## Conclusions

The current study demonstrated that the complete-arch models manufactured with the tested stones were within clinical tolerance and could be appropriate for the production of fixed restorations. The tested CAD/CAM optimized complete-arch stone models (BC and EM) were not superior to type IV conventional stone models (ERF) in terms of trueness. However, resin-reinforced scannable stone (EM) was significantly more precise than other tested stones. The resin-reinforced scannable stone (EM) showed a much denser point cloud compared to the other stones evaluated.

## Supporting information

**S1 Dataset.**
(XLSX)

## Author Contributions

**Conceptualization:** Gülsüm Ceylan, Faruk Emir.

**Data curation:** Faruk Emir.

**Formal analysis:** Faruk Emir.

**Investigation:** Gülsüm Ceylan.

**Methodology:** Gülsüm Ceylan, Faruk Emir.

**Validation:** Gülsüm Ceylan, Faruk Emir.

**Visualization:** Faruk Emir.

**Writing – original draft:** Gülsüm Ceylan, Faruk Emir.

**Writing – review & editing:** Gülsüm Ceylan, Faruk Emir.

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
