## [Decision Letter · Decision Letter 0]

7 Feb 2023

PONE-D-22-35383Evaluating the Accuracy of CAD/CAM Optimized Stones Compared to Conventional Type IV StonesPLOS ONE

Dear Dr. Ceylan,

Thank you for submitting your manuscript to PLOS ONE. After careful consideration, we feel that it has merit but does not fully meet PLOS ONE’s publication criteria as it currently stands. Therefore, we invite you to submit a revised version of the manuscript that addresses the points raised during the review process.

Please note the comments of the reviewers and address the concerns raised. I look forward to receiving the revised manuscript. 

We look forward to receiving your revised manuscript.

Kind regards,

Mirza Rustum Baig

Academic Editor

PLOS ONE

Journal Requirements:

"NO"

Reviewers' comments:

Reviewer's Responses to Questions

**Comments to the Author**

1. Is the manuscript technically sound, and do the data support the conclusions?

Reviewer #1: Yes

Reviewer #2: Yes

2. Has the statistical analysis been performed appropriately and rigorously? 

Reviewer #1: Yes

Reviewer #2: Yes

3. Have the authors made all data underlying the findings in their manuscript fully available?

Reviewer #1: Yes

Reviewer #2: Yes

4. Is the manuscript presented in an intelligible fashion and written in standard English?

Reviewer #1: Yes

Reviewer #2: Yes

5. Review Comments to the Author

Reviewer #1: Comments

This is a simple study but useful in clinical application. Some comments.

In this statement “Different devices such as analog or digital calipers, profile projectors, and microscopes have been to evaluate the accuracy of stone casts.” Please add following references.

https://doi.org/10.3390/ma15093004

https://doi.org/10.32604/cmc.2020.011943

Method.

Add details on the printed master model. Software used and processing.

Please discuss using following articles.

Ambient Lights can affect the Accuracy of a 3-Dimensional Optical Scanner. https://doi.org/10.1155/2022/2637078

Please discuss with previous similar studies. https://doi.org/10.1155/2021/2673040

Conclusion

It is not necessary to add P value in the conclusion. Pleas edit and present in a better way.

Reviewer #2: This is an interesting study evaluating the accuracy of CAD/CAM-optimized stones compared to conventional type IV stones. Hence, the study adds to the current knowledge in this field and is definitely of clinical significance. However, the following matters are still unclear to me as far as I read the manuscript.

Abstract

Definitions of each group are missing in the abstract. Adding a description, including an abbreviation for each group, is recommended.

Materials and Methods

Alignment between each pair of scan data was superimposed using the best-fit algorithm, but I'm afraid I have to disagree with the author's method. Suppose there is deformation across the complete arch due to the expansion and contraction of the stone model. In that case, the total expansion/contraction can only be evaluated by aligning the scan data pair to the local area. For example, if the authors used the 'local best-fit alignment' aligning only on A1, the deviation at A6 could have shown the expansion/contraction of the entire model. The authors evaluated the complete arch model, but it should be explained why they aligned comparing data with the best-fit algorithm, a method in which the deformity components are canceled out on average by relying on simple computer calculations.

Result

When the point cloud density was compared, there was only a difference of 6 points at most per 1 square millimeter. It is doubtful whether this has any clinical implications. Deviations were measured at different locations from A1 to A6. Still, the information in table 5 shows just the reduced deformation of the complete arch model. There is no information on how each cylinder from A1 to A6 is deformed in the XYZ direction. Therefore, the authors and readers can only evaluate the amount of deformation of models made of each material with limited information.

Discussion

In the text, the author said that each abutment was analyzed separately. However, it was not explained whether the alignment was conducted independently. I cannot discriminate whether the RMS was calculated on each cut abutment after alignment and leaving the positional relationship as it is or not. I guess the alignment was conducted while the abutments were grouped, and only the RMS evaluated each abutment separately. Even so, the evaluation method of this study still provides only limited information to assess the expansion/contraction patterns that occur during impression taking and plaster setting.

The results of this study were directly compared with the marginal gap of the prosthesis, but this is not appropriate. This is because the abutment itself may have moved due to the expansion/contraction of the entire arch, and such deviations are added.

Conclusion

Likewise, it is not correct to conclude by directly comparing the results of this study with clinically acceptable marginal gaps. Also, although there is a statistical difference in point cloud density, it is questionable whether this is a clinically meaningful conclusion.

6. PLOS authors have the option to publish the peer review history of their article (what does this mean?). If published, this will include your full peer review and any attached files.

Reviewer #1: No

Reviewer #2: **Yes: **Ji-Man Park

---

## [Author Response · Author response to Decision Letter 0]

14 Feb 2023

Dear Editor,

We appreciate your allowing us to revise and resubmit our paper “Evaluating the Accuracy of CAD/CAM Optimized Stones Compared to Conventional Type IV Stones.” 

We have made all the changes for PLOS ONE's style requirements. In addition, as you have stated, we have included the necessary explanation about sources of funding (financial or material support) for our study in the Cover Letter. We also added an explanation about the Data Availability statement to the content of the Cover Letter. Finally, we have added the data of our study to the Supporting Information section as S1_Dataset.

We sincerely appreciate the reviewer's comments and suggestions, which have greatly helped improve the manuscript's quality. Therefore, we have corrected and modified the manuscript according to the comments given by the reviewers. 

The authors have approved the revisions. The changes are highlighted in yellow in the original paper. The responses are given below.

Reply to Reviewer #1

1. Concern of the reviewer: In this statement “Different devices such as analog or digital calipers, profile projectors, and microscopes have been to evaluate the accuracy of stone casts.” Please add following references.

https://doi.org/10.3390/ma15093004

https://doi.org/10.32604/cmc.2020.011943

Our response: Explanations and references were added based on recommendations in the Discussion section.

Revised text: Different devices such as analog or digital calipers, profile projectors, and microscopes have been to evaluate the accuracy of stone casts [26,27].

• The references listed below:

26. Thangwarawut P, Amornvit P, Rokaya D, Kiattavorncharoen S. Comparison of Different Types of Static Computer-Guided Implant Surgery in Varying Bone Inclinations. Materials (Basel). 2022;15(9): 3004. (https://doi.org/10.3390/ma15093004)

27. Amornvit P, Rokaya D, Peampring C, Sanohkan S. Confocal 3D Optical Intraoral Scanners and Comparison of Image Capturing Accuracy. Comput Mater Contin. 2021;66(1): 303-314. (https://doi.org/10.32604/cmc.2020.011943)

2. Concern of the reviewer: (Method) Add details on the printed master model. Software used and processing.

Our response: We thank the reviewer for his valuable contributions. Requested details are given in the Materials and methods section.

Revised text: The digital master model was then saved in Standard Tessellation Language (.stl) format and printed using a Polyjet 3D printer (Objet30 Prime, Stratasys Ltd.). The Polyjet 3D printers utilize materials that are extruded from nozzles or a photopolymer that is jetted over the workspace. Then the object is solidified through polymerization with the use of a UV light source [17]. The Objet30 Prime printer uses the intuitive “Objet Studio” 3D printing software. A rigid and transparent photopolymer material named VeroClear was used to print the master model, while the SUP705 material was used for support structures. The SUP705 material is removable with a waterjet, so no post-curing process was necessary. The production time of the master model took 4 hours and 10 minutes.

3. Concern of the reviewer: Please discuss using following articles. “Ambient Lights can affect the Accuracy of a 3-Dimensional Optical Scanner.”

https://doi.org/10.1155/2022/2637078

Please discuss with previous similar studies. 

https://doi.org/10.1155/2021/2673040

Our response: The following sentences and references are added to the Discussion section based on recommendation. 

Revised text: 

• Thongma-Eng et al. [41] reported that the color of the scanned object influenced the accuracy of the scans. Therefore, the colors of the plaster models were selected to be similar to each other.

• Some studies have reported that the accuracy of intraoral scans decreases with the increased length of the scanned model, which can result in registration problems of the overlapping images [14,28-34]. 

• Objects with smaller dimensions can be digitized more accurately than larger ones, such as the complete dental arch [12,34].

• The references listed below:

34. Amornvit P, Rokaya D, Sanohkan S. Comparison of Accuracy of Current Ten Intraoral Scanners. Biomed Research International 2021;2021: 2673040.

41. Thongma-Eng P, Amornvit P, Silthampitag P, Rokaya D, Pisitanusorn A. Effect of Ambient Lights on the Accuracy of a 3-Dimensional Optical Scanner for Face Scans: An In Vitro Study. J Healthc Eng. 2022;2022: 2637078.

4. Concern of the reviewer: (Conclusion) It is not necessary to add p value in the conclusion. Please edit and present in a better way.

Our response: It is corrected accordingly.

Revised text: The current study demonstrated that the complete-arch models manufactured with the tested stones were within clinical tolerance and could be appropriate for the production of fixed restorations. The tested CAD/CAM optimized complete-arch stone models (BC and EM) were not superior to type IV conventional stone models (ERF) in terms of trueness. However, resin-reinforced scannable stone (EM) was significantly more precise than other tested stones. The resin-reinforced scannable stone (EM) showed a much denser point cloud compared to the other stones evaluated.

Reply to Reviewer #2

1. Concern of the reviewer: (Abstract) Definitions of each group are missing in the abstract. Adding a description, including an abbreviation for each group, is recommended.

Our response: The description, including the abbreviations, has been added to the Abstract section, based on the recommendation.

Revised text: This study compared the accuracy (trueness and precision) of stone models fabricated using two brands of CAD/CAM optimized stones Cerec Stone (BC) and Elite Master (EM), and a conventional type IV stone Elite Rock Fast (ERF).

2. Concern of the reviewer: (Materials and Methods) Alignment between each pair of scan data was superimposed using the best-fit algorithm, but I'm afraid I have to disagree with the author's method. Suppose there is deformation across the complete arch due to the expansion and contraction of the stone model. In that case, the total expansion/contraction can only be evaluated by aligning the scan data pair to the local area. For example, if the authors used the 'local best-fit alignment' aligning only on A1, the deviation at A6 could have shown the expansion/contraction of the entire model. The authors evaluated the complete arch model, but it should be explained why they aligned comparing data with the best-fit algorithm, a method in which the deformity components are canceled out on average by relying on simple computer calculations.

Our response: In our study, we used best-fit alignment for evaluating complete-arch models. Before the evaluation process of abutments (A1 to A6), each abutment was separated from the master model and saved as separate data. Afterwards, a 3D comparison (best-fit alignment method used) was performed between each individual abutment and the corresponding abutment on the master model. For this reason we don’t use a local best fit algorithm. 

Figure 1. Shows the separated abutment .stl data.

Figure 2-4. Show the data of the individual abutments (A1 to A6) for the EM, ERF, and BC models.

Figure 5. Shows a 3D comparison report between the A1 abutment of the BC-1 stone model and the A1 abutment of the master model. Each abutment was compared individually to its corresponding abutment on the master model.

3. Concern of the reviewer: (Result) When the point cloud density was compared, there was only a difference of 6 points at most per 1 square millimeter. It is doubtful whether this has any clinical implications. 

Our response: We changed this sentence “The point cloud density of the resin-reinforced scannable stone (EM) was significantly higher than the other tested stones.” to “The resin-reinforced scannable stone (EM) showed a much denser point cloud compared to the other stones evaluated.” As written in discussion section all models were clinically acceptable.

4. Concern of the reviewer: (Result) Deviations were measured at different locations from A1 to A6. Still, the information in table 5 shows just the reduced deformation of the complete arch model. There is no information on how each cylinder from A1 to A6 is deformed in the XYZ direction. Therefore, the authors and readers can only evaluate the amount of deformation of models made of each material with limited information.

Our response: We concur with the reviewer regarding this matter. The lack of representation of deviations in the X, Y, and Z directions in the models is a noted limitation in our study and has been acknowledged in the final sentence of the “Discussion" section. 

This sentence “The present study has the following limitations: 3D analysis was performed using standardized models in vitro, and only one impression material and dental scanner were used. Future studies should include the surface roughness of the stone models, different brands of CAD/CAM optimized stones, and calculation of distortions in the X, Y, and Z directions.” is changed based on your valuable recommendation.

Revised text: The limitations of the present study include: the 3D analysis was conducted in vitro using standardized models, only one impression material and dental scanner were utilized, and the deviations in the X, Y, and Z directions were not determined. Future studies should include the surface roughness of the stone models and different brands of CAD/CAM optimized stones.

5. Concern of the reviewer: (Discussion) In the text, the author said that each abutment was analyzed separately. However, it was not explained whether the alignment was conducted independently. I cannot discriminate whether the RMS was calculated on each cut abutment after alignment and leaving the positional relationship as it is or not. I guess the alignment was conducted while the abutments were grouped, and only the RMS evaluated each abutment separately. Even so, the evaluation method of this study still provides only limited information to assess the expansion/contraction patterns that occur during impression taking and plaster setting.

Our response: In this study, best-fit alignment was used to assess complete-arch models. The evaluation of each abutment (A1 to A6) was conducted by first separating it from the master model and saving it as individual data. Then, a 3D comparison (best-fit alignment method used) was carried out between each individual abutment and the corresponding abutment on the master model. For this reason we don’t use a local best fit algorithm. 

According to Rudolph et al.* “F or the RMS-error, all positive and negative values are first squared, then the mean is calculated and finally the root is extracted thus preventing positive and negative values to neutralize each other.” If the positive and negative deviations in the data have an equal distribution, the average values will be close to zero. The RMS value can be a better measure of accuracy than the average of the positive and negative deviations. The RMS value takes into account both the positive and negative deviations and gives a more representative measure of the overall deviation. In contrast, the average of the positive and negative deviations will be close to zero if the positive and negative deviations are equally distributed. This can give a misleading indication of the accuracy of the data, as the overall deviation may still be significant even if the average of the positive and negative deviations is close to zero. For data with both negative and positive numbers, the mean or average can be misleading, as it can be skewed by positive or negative values. In these cases, RMS can be a better measure.

*Rudolph H, Graf MR, Kuhn K, Rupf-Köhler S, Eirich A, Edelmann C, Quaas S, Luthardt RG. Performance of dental impression materials: Benchmarking of materials and techniques by three-dimensional analysis. Dent Mater J. 2015;34(5):572-84.

6. Concern of the reviewer: (Discussion) The results of this study were directly compared with the marginal gap of the prosthesis, but this is not appropriate. This is because the abutment itself may have moved due to the expansion/contraction of the entire arch, and such deviations are added.

Our response: We appreciate the reviewer for his valuable contribution. The accuracy of the stone models, scanners, and framework, and the type of cement or cementation method used, all play a critical role in determining the marginal gap. Given that the total errors in all of these factors can impact the final prosthesis marginal gap, it is important that the accuracy of the stone models is below the clinically accepted marginal gap values. In this study, despite the master model having specific landmarks, each abutment was isolated and analyzed individually to remove any impact from deviations caused by scanning a large span, which can lead to arch distortion.

7. Concern of the reviewer: (Conclusion) Likewise, it is not correct to conclude by directly comparing the results of this study with clinically acceptable marginal gaps. Also, although there is a statistical difference in point cloud density, it is questionable whether this is a clinically meaningful conclusion.

Our response: We thank the reviewer for his important contributions. The accuracy of the stone or printed models, the accuracy of the scanners, and the substructure of the prosthesis, as well as the type of cement or cementation technique used, are all contributors to the marginal gap. Since the sum of errors in all factors will affect the final prosthesis marginal gap. For this reason the accuracy of stone models should be below the accepted marginal gap values. The results section has been written in reference to the commonly accepted clinical limits used in many studies.

Note to editor: We have added new references to the reviewers' recommendations. Therefore, the references section has been changed. 

We hope to have addressed the reviewer’s concerns satisfactorily.

Sincerely

---

## [Editor Report · Decision Letter 1]

17 Feb 2023

Evaluating the Accuracy of CAD/CAM Optimized Stones Compared to Conventional Type IV Stones

PONE-D-22-35383R1

Dear Dr. Ceylan,

We’re pleased to inform you that your manuscript has been judged scientifically suitable for publication and will be formally accepted for publication once it meets all outstanding technical requirements.

Kind regards,

Mirza Rustum Baig

Academic Editor

PLOS ONE

Additional Editor Comments (optional):

Please add the number of abutments used in the master model in the 'Abstract' section.
---

## [Editor Report · Acceptance letter]

23 Feb 2023

PONE-D-22-35383R1 

Evaluating the Accuracy of CAD/CAM Optimized Stones Compared to Conventional Type IV Stones 

Dear Dr. Ceylan:

I'm pleased to inform you that your manuscript has been deemed suitable for publication in PLOS ONE. Congratulations! Your manuscript is now with our production department. 

Kind regards, 

on behalf of

Dr. Mirza Rustum Baig 

Academic Editor

PLOS ONE